# Chemical Structures, Properties, and Applications of Selected Crude Oil-Based and Bio-Based Polymers

**DOI:** 10.3390/polym14245551

**Published:** 2022-12-19

**Authors:** Piotr Koczoń, Bartłomiej Bartyzel, Anna Iuliano, Dorota Klensporf-Pawlik, Dorota Kowalska, Ewa Majewska, Katarzyna Tarnowska, Bartłomiej Zieniuk, Eliza Gruczyńska-Sękowska

**Affiliations:** 1Department of Chemistry, Institute of Food Sciences, Warsaw University of Life Sciences, 02-776 Warsaw, Poland; 2Department of Morphological Sciences, Institute of Veterinary Medicine, Warsaw University of Life Sciences, 02-776 Warsaw, Poland; 3Faculty of Chemistry, Warsaw University of Technology, 00-664 Warsaw, Poland; 4Department of Food Quality and Safety, Poznan University of Economics and Business, 61-875 Poznan, Poland

**Keywords:** crude oil-based, bio-based polymers, chemical structure, properties, application

## Abstract

The growing perspective of running out of crude oil followed by increasing prices for all crude oil-based materials, e.g., crude oil-based polymers, which have a huge number of practical applications but are usually neither biodegradable nor environmentally friendly, has resulted in searching for their substitutes—namely, bio-based polymers. Currently, both these types of polymers are used in practice worldwide. Owing to the advantages and disadvantages occurring among plastics with different origin, in this current review data on selected popular crude oil-based and bio-based polymers has been collected in order to compare their practical applications resulting from their composition, chemical structure, and related physical and chemical properties. The main goal is to compare polymers in pairs, which have the same or similar practical applications, regardless of different origin and composition. It has been proven that many crude oil-based polymers can be effectively replaced by bio-based polymers without significant loss of properties that ensure practical applications. Additionally, biopolymers have higher potential than crude oil-based polymers in many modern applications. It is concluded that the future of polymers will belong to bio-based rather than crude oil-based polymers.

## 1. Introduction

The increasing demand for crude oil-based materials has contributed to the depletion of natural reserves of petroleum. It is estimated that a serious shortage in crude oil and significant increase in its costs will be noticed as early as 2040 [1], and petroleum supplies will have been consumed until the end of this century. As much as 11–12% of crude oil is used in the production of polymers [2,3]. Fossil fuel-based polymeric materials show a variety of desirable physical properties such as durability, light weight, and resistance to corrosion and chemical reagents. For this reason the spectrum of the applications of crude oil-based polymers is extremely wide, from packaging [4] to constructional materials [5] and medical equipment [6]. Unfortunately, the properties that make fossil-based polymers so attractive trigger enormous environmental problems, since the vast majority of petroleum-based polymers do not decompose and continue to remain almost untouched for centuries. Environmental issues along with the upcoming deficiency in crude oil supplies have gained a great concern among the polymer industry and researchers who try to find new alternatives to crude oil-based polymers. Polymers from renewable resources have recently attracted the attention of global scientists as a future perspective in replacing crude oil-based polymers or reducing their usage.

Bio-based polymers are polymers derived (at least in part) from renewable raw materials such as plant and animal biomass, and organic waste. This group of materials can be biodegradable (such as polylactic acid—PLA) or nondegradable (such as biopolyethylene—Bio-PE). However, bio-based polymers are considered environmentally responsible, since they do not depend on finite fossil fuel reserves and can be obtained with a lower carbon footprint than their crude oil counterparts. Renewable raw materials such as starch, vegetable oils, and proteins are nearly inexhaustible natural resources, the supply of which can be restored in a short period of time. Renewables are inexpensive and readily available, ensuring sustainable carbon transfer from biomass and similar materials to bio-based polymers or intermediates used in their production.

Natural bio-based polymers of industrial application can be directly extracted from biomass. Such polymers are produced in large quantities by plants (e.g., cellulose, hemicellulose, starch, inulin, and pectin) or by animals (e.g., chitin and chitosan), and they often need to undergo special modification to meet requirements for their use [7]. Molecular biology and genetic engineering can be successfully incorporated into the process of creating agricultural crops of desirable properties or facilitating the procedure for subsequent recovery of biopolymers. Bio-based polymers can also be synthesized de novo by various bacterial strains (e.g., *Pseudomonas putida*, *Aeromonas hydrophila*, *Bacillus subtilis*) during fermentation processes using low molecular weight metabolism products. Another group of bio-based polymers includes synthetic polymers from bio-derived monomers (e.g., PLA and other polyesters). Several prominent building blocks such as succinic, itaconic, muconic, and lactic acids can be effectively produced from biomass [8].

The physicochemical properties of bio-based polymers are often similar to crude oil-based polymers, which makes them a potential substitute for their crude oil-based counterparts. Moreover, bio-based polymers can exhibit a wide range of new features, which can enable novel applications in many technological fields. Not only should bio-based production be cost-effective but also more sustainable in comparison to crude oil-based production.

The paper discusses the possibility of replacing selected popular crude oil-based polymers such as polyethylene, low density polyethylene, polystyrene, polyethylene terephthalate, and polyvinyl chloride with bio-based polymers made from renewable resources such as polylactic acid (PLA), derived from an animal origin, such as chitosan, and received on biotechnological pathways such as polyhydroxybutyrate (PHB) and pullulan. The choice of the petroleum-based polymers was dictated by their versatile application in a multitude of industries, whereas PLA and PHB are the major industry players that are bringing bio-based polymeric materials to the market, and polysaccharides such as pullulan and chitosan are fully appreciated by researchers and industrialists worldwide for their high biocompatibility and biodegradability. The properties and practical applications of both groups of polymers have been collected in order to compare them and indicate the possible future perspectives for bio-based polymers.

## 2. Polyethylene vs. Polyhydroxybutyrate

Shopping bags, including those from supermarkets, plastic packaging, various lids, industrial foils, milk containers, and others, including thin-wall containers, wire insulators, pipes, injection, and blow molding, are all made from different types of chemically the same material—polyethylene (PE). In addition to the products listed that are in common use, clearly visible in modern world, PE has many other important, yet not so commonly known applications, e.g., parts of various machines used in the food and paper industry, veterinary and medicine tools and materials, and in the construction industry [9]. Although there are currently many voices pointing out PE disadvantages, above all its inability to be naturally degradable and having a share in the dramatic increase in the volume of litter, one must admit that PE is a companion in everyday life [10].

The history of PE can be considered serendipitous both in terms of laboratory discovery and industrial synthesis. It goes back to 1898 when Hans von Pechmann accidentally obtained a white waxy substance by heating diazomethane. Later on in 1900, a similarly obtained waxy substance was proved to contain a long carbon chain and was given the name “polymethylene” by Eugen Bamberger and Friedrich Tschirner. In both cases, the waxy substance was polyethylene. Laboratory discovery was followed by industrial synthesis of PE at the works of Imperial Chemical Industries (ICI), Northwich, the UK, in 1933. The very first patent referring to production of PE from ethylene was granted on 6 September 1937. Between first information in 1898 and the patent in 1937, several reactions were tested to produce the waxy substance first obtained by Pechmann. Numerous chemical pathways to obtain PE were worked out, which included several reactions, e.g., reaction of decamethylene dibromide with sodium in a Wurtz-type reaction done by Carothers and Van Natta in 1930 [11], hydrogenation of polybutadiene, modified Fischer–Tropsch reduction of carbon monoxide with hydrogen, or reduction of poly(vinyl chloride) with lithium aluminum hydride. Every pathway has advantages and disadvantages, e.g., inability to obtain a polymer with molecular mass greater than 1300 with use of Wurtz-type reaction or formation of branched-only or unbranched-only products. Currently, ethylene and other materials generated from crude oil are primary source materials to produce PE commercially [12,13,14].

The monomer of PE, ethylene, is chemically unsaturated and contains a double bond, while PE is saturated containing only single bonds between carbon atoms (Figure 1). Carbon chains can be straight or branched. In general, if the chains are straight, the basic form of PE is considered, while branched chains are associated with low-density PE (LDPE). Both forms are produced in polymerization reactions taking place under high pressure and temperature. Ethylene is heated and pressed to break double bonds and form long chains. Pressure is applied to make less space for ethylene molecules that are in gaseous form. Although no equilibrium is considered in the polymerization reaction, the Le Chatelier principle can be used to explain favoring of the forward direction: solid state products (right-hand side of reaction) occupy much less space than gaseous substrates (left-hand side of reaction) [12,15].

The production process starts with preparation of ethylene with appropriate properties. The monomer is sourced by dehydration of ethyl alcohol fermented from molasses. Another source for ethylene is a selected fraction of crude oil fractional distillation. Before starting the polymerization process, all undesired contaminants such as carbon monoxide, oxygen, water, and acetylene must be removed. The use of inadequately prepared reactant yields product of undesired properties, especially in terms of insulation or heat resistance [10,12].

The polymerization route requires applying high pressure of 100–300 MPa and temperature of 350–600 K. This is the most common process, known as high-pressure polymerization. There are also other well-known technological methods of PE production, e.g., Standard Oil Company Indiana process, high-density PE (HDPE) Phillips or Ziegler process, and metallocene catalyzed process. Application of different conditions, namely pressure, temperature, or catalyst, leads to production of PE, LDPE, or HDPE [10,12,16].

Currently PE is divided into classes that consider chemical structure and resultant properties [10,12]. Structural classes are:Linear PE—with a macromolecule made of many monomer units arranged in a straight line;Cross-linked PE (PEX)—where macromolecule has covalent bonds between the polymer molecules.

PE can also be classified according to material density. The following classes are described:Low density PE (LDPE);Linear low density PE (LLDPE);Middle density PE (MDPE);High density PE (HDPE);Ultra-high molecular weight PE (UHMWPE).

In summary, PE with a general molecular formula of –[C_2_H_4_]_n_–, a degree of polymerization of 1500–9000, and a melting point at 395–400 K [15] has very useful and desirable mechanical and chemical properties, including being soft, transparent, tasteless and odorless, but not resistant to high temperature. The main advantage of all PE types is low price, as costs of production are low. PE from each class has different properties; hence, it is used in different areas of human life, for example, LLDPE has the highest impact strength, tensile strength, and extensibility of all PE classes [15,17,18,19].

Without specific treatment, decomposition of PE takes hundreds of years, hence many scientific and industrial interdisciplinary teams work on methods to increase its decomposition rate or reuse [20,21,22,23]. Promising investigations cover the use of bacteria, yeasts, and enzymes present in microorganisms to recover material that can be used for the next synthesis of plastic exhibiting the quality equal to the one obtained in petrochemical processes. Investigated microorganisms can also be used for remediation of plastic waste present in soil and landfills [24,25,26]. However, there is no similar research on PE/LDPE biomedical wastes.

Polyhydroxybutyrate—PHB—was discovered in 1925 by Maurice Lemoigne [27,28]. Chemically, it is polyester. There are several slightly different forms of this polymer, namely poly-4-hydroxybutyrate (P4HP), polyhydroxyvalerate (PHV), polyhydroxyhexanoate (PHH), and the most common poly-3-hydroxybutyrate (P3HB).

With the molecular formula [OCH(CH_3_)CH_2_CO]_n_ (Figure 2), PHB has similar mechanical properties to PE, while its greatest advantage over PE is biodegradability [29]. However, the biggest disadvantage is its high cost of production, and currently there is intensive research focused on decreasing these costs [24,30,31,32].

PHB that belongs to a polyhydroxyalkanoate (PHA) family is a semicrystalline thermoplastic polyester. Its glass transition temperature is 278–282 K; the melting point is 440–450 K. PHB has a very low Young’s modulus (3–3.5 GPa) compared to other biodegradable biopolymers. Its decomposition lasts up to several years [15,33]. PHB is used as a component in medical implants, surgical sutures, and elements of artificial tissues [34,35].

PHB can be obtained from its monomer—butyric acid—by microorganisms on biological pathways [29]. Production of PHB is more expensive than any other PE. Therefore, sources of cheap organic matter are required. One way is food-originated residuals that undergo fermentation to form volatile fatty acids, which are in turn converted by specific bacteria, e.g., *Cupriavidus necator* to form PHB. Another method of PHB production is the use of the strain *R. piridinivorans* BSRT1-1. This strain is isolated from soil, and it has been stated that fructose and KNO_3_ are the best sources of carbon and nitrogen for bacteria to produce PHB. Under optimal conditions, 3.60 g of PHB can be formed from 1 dm^3^ of biomass. Details are provided in [36]. *Bacillus cereus* SH-02 (OM992297) is also considered to be a good producer of PHB [37].

Both PE/LDPE and PHB are widely used in medicine and the veterinary and agricultural industries. Although both are polymers, their physical, chemical, and biological properties, together with their formation and methods of utilization, differ significantly. Both plastics are commonly used in production of medicinal and veterinary materials [12,29,38,39]. Materials and products obtained from both polymers can be in direct touch with human organs. They can be used for production of bone tunnels [40], surgical threads [41], containers, foils, slices, packaging [42], and antibacterial foils [43]. Owing to relatively low costs, PE single-use products are manufactured and applied [21]. Orthopedics is a rapidly developing area for the use of polymers, including PE and PHB applications. In the most common surgery worldwide, i.e., total knee arthroplasty (TKA), endoprostheses used are made of PE. A specific type of PE used for this type of endoprostheses provides physiological kinematics of the knee joint [44,45]. Additional advantage is low specific weight, mechanical resistance, and transparency to X-rays [46].

A characteristic and practical important feature of this biopolymer is its significantly high biodegradability and biocompatibility. As biopolymers generally, PHB can be relatively easily decomposed by enzymes produced by living species. In the biological environment, biodegradation can occur by oxidation, including photo-oxidation or hydrolysis [47]. PHB can be easily decomposed inside human tissues by enzymatic or hydrolytic decomposition, with pH kept at a constant level [48,49]. Products formed are easy metabolized by human or animal organisms [50]. This ability to be degradable, especially in the human body, allows use of PHB as a composite for implants. Additionally, PHB can promote cell growth in a near-natural biological environment [28,49,51,52].

## 3. Polystyrene vs. Polylactide

Polystyrene (PS) is a thermoplastic polymer (Figure 3) made of aromatic hydrocarbon monomer styrene that is derived from fossil-fuels [53].

The synthesis of PS is based on the free radical polymerization of styrene using free-radical initiators. It is mostly used in solid (high impact and general purpose PS), foam and expanded PS forms. The main advantages of PS are low-cost, easy processing ability, and resistance to ethylene oxide, as well as radiation sterilization. It is, however, not resistant to organic solvents such as cyclic ethers, ketones, acids, and bases. The most popular general purpose PS (GPPS or unmodified PS) is transparent, brittle, and rigid, which makes this kind of material suitable for laboratory purposes, such as diagnostic and analytical, and medical packaging (e.g., Petri dishes, tissue culture trays, pipettes, test tubes). For high-strength products, high-impact PS (HIPS) is competitive with polypropylene and PVC [54]. It is typically used in thermoformed products, such as catheters, heart pumps, and epidural trays, and toys, packaging, and electronic appliances. Owing to its high dimensional stability and easy processing, it is often chosen for the preproduction prototypes in 3D-printing technique [55]. As a result of strong C-C and C-H bonds present in the structure, PS is resistant to biodegradation without special treatment such as copolymerization and fictionalization. However, it was proved that some bacterial species are able to form biofilm on the PS surface, which leads to its partial degradation [56]. PS can be recycled using several methods. Mechanical recycling is the one with lowest cost, but it has many limitations. The main obstacle is efficient separation of PS from the plastic waste stream. Currently, PS is sorted using near-infrared technologies and complementary sorting methods, including density, electrostatics, selective dissolution, and flotation [57]. The latter—froth flotation—is the most common due to its low cost and the possibility to separate polymers with similar density. To increase the flotation effectiveness, surface modification can be performed, e.g., in the presence of KMnO_4_ [58] or K_2_FeO_4_ [59]. Recycled PS exhibits worse mechanical properties than neat polymer, and reduction in molecular mass is also observed. Nevertheless, many products made of recycled PS can be found on the market, e.g., pencils, doors, window frames, cups, plates, and bottles; some of them even approved for food contact [60]. Chemical recycling of PS is less common due to the high cost. It leads to the production of styrene, and other useful chemicals such as benzene, toluene, indan, ethylbenzene, and benzoic acid, via pyrolysis and oxidation. Recently, a novel simple and low-cost method has been reported that enables the oxidative cleavage of PS to benzoic acid, formic acid, and acetophenone by singlet oxygen at ambient temperature and pressure [61]. PS waste can also be converted to biodegradable PHAs [62]. To summarize, PS is one of the most important polymers present in our daily life. However, once it has fulfilled its designed purpose, it is not easily degradable. Chemical recycling is not economically convenient since the feedstocks are cheaper than the process itself; additionally, mechanical recycling is limited due to the low separation efficiency of PS from the plastic waste stream. That is why there is an urgent need to find sustainable alternatives that can at least partially replace petroleum-based PS in use. The most popular green substitutes for PS are cellulose and thermoplastic starch used as thermal insulation materials (foams) [63,64], and poly(vinyl alcohol) for bead-foaming process [65] and polylactide [66].

Polylactide (PLA)—biodegradable and compostable aliphatic polyester (Figure 4)—is one of the key biopolymers with the largest market significance. The global volume of PLA production was around 457,000 metric tons in 2021, which accounted for 29% of the total biodegradable bioplastics production worldwide [67]. The PLA production on industrial scale is either based on the ring-opening polymerization (ROP) of lactide, (method applied by NatureWorks LLC, Plymouth, MN, United States, and Corbion N.V., Amsterdam, the Netherlands) or direct polycondensation of lactic acid in an azeotropic solution (applied by Mitsui Toatsu Chemicals, Inc., Tokyo, Japan) [68]. In both cases, high molecular mass PLA is obtained; however, solvent-free ROP is preferable for production in large scale. In this case, optically pure *L*-lactic or *D*-lactic acid is produced as a monomer of PLA by microbial fermentation from renewable resources such as molasses, whey, sugar cane, and plants with high starch content [69]. Next, LA is condensed to form low molecular mass prepolymer PLA, which undergoes a controlled depolymerization to a cyclic dimer of lactate–lactide. The polymerization of lactide is generally catalyzed by tin octanoate and requires short reaction time at a temperature of about 440–460 K [70].

The mechanical properties of PLA are similar to those of PS and polyethylene terephthalate (PET) [71], also described in detail in Section 4 of this paper. It can also be a sustainable alternative to polypropylene (PP) and PVC. PLA is as rigid and brittle as PS, and its resistance to fats and oils resembles PET [71]. Although CO_2_, O_2_, N_2_, and H_2_O permeabilities for PLA are higher than for PET, but lower than for PS [72], therefore many attempts to improve the PLA barrier properties have been reported, e.g., by introducing nanofillers with a lamellar structure [73]. In addition, it is characterized by a high tensile modulus and resistance to UV radiation. Good mechanical and optical properties allow PLA to compete with the existing crude oil-based thermoplastics. PLA containing approx. 5% of *D*-repeating units is a transparent, colorless, and relatively rigid material resembling PS [74]. An extra advantage of PLA is its easy processing ability through conventional melt processes such as extrusion, injection molding, compression molding, or blow molding, which are also used for other commercial polymers, namely PS and PET [75].

The properties of PLA depend on the polymer molecular mass and the degree of crystallinity [76]. Stereochemistry also plays an important role. The stereochemical composition and distribution of monomer units along the polyester chain affect the properties of PLA [74]. *L*-PLA (PLLA) and *D*-PLA (PDLA) are composed of lactic acid units of the same chirality [77]. They are isotactic, stereoregular, and partially crystalline polymers (degree of crystallinity up to 60%), the glass-transition temperature (T_g_) is approx. 320–330 K, and the melting point is 440–470 K [29,74,78]. On the other hand, *D*,*L*-PLA is an amorphous polymer with a T_g_ of about 330 K. It shows worse mechanical properties and degrades faster than PLLA and PDLA. The highest melting point, about 500 K, shows a racemic mixture of PLLA and PDLA, in which chains of different chirality form a densely packed network. Compared to the parent polymers, the resulting racemic PLA (PDLLA) has enhanced functional properties, such as mechanical strength, durability, and thermal and hydrolytic stability [79].

Desirable properties allow PLA to compete with PS in several application fields, described below.

### 3.1. Packaging Application

Low toxicity, strong flavor and aroma barrier, and high transparency make PLA an ideal material for fresh food packaging, especially fruit and vegetables [80]. Auras et al. [81] tested and compared oriented PLA (OPLA) with PET and oriented PS (OPS) films intended for production of fresh fruit and vegetables storage containers. According to these results, mechanical, physical, and barrier properties of OPLA were comparable and, in some cases, better than standard OPS and PET containers. Similar studies were performed for the shelf life of blackberries [82] and blueberries [83] under retail conditions closed in the OPS and OPLA containers. In both cases the shelf life was extended, proving that PLA can be a good replacement for PS. PLA can be used also as trays for storage of mangoes, melons, and other tropical fruit. The shelf life of the fruit packed in such a way was the same as of the fruit packed in PET trays [84]. However, the PLA packaging is more susceptible to cracking and breaking during transport when compared with OPS or PET. Neither the sheet nor the finished product can be stored at temperatures above 313 K or relative humidity greater than 50% [85].

### 3.2. Three-Dimensional Printing

The filaments used in 3D printing are primarily thermoplastics. The most popular are PLA, acrylonitrile butadiene styrene (ABS) and HIPS [86]. In all three cases, filament can also be produced from recycled plastic, which can significantly reduce its price. It is worth mentioning that commercial filaments for 3D printing are 20 to 200 times more expensive than those of raw plastics [87]. The source for PLA waste is food containers and bottles, ABS filaments originating from car dashboards, and HIPS derived from refrigerators or automotive parts [88]. The advantages of PLA as filament for 3D printing are ease of printing, glossiness, and multicolor appearance. The dimensional accuracy of the parts printed from PLA is high since it poses less warp behavior than the other filaments. Compared to HIPS, PLA filament does not require a heated bed, it is odorless, and what is more important, it releases many fewer volatile organic compounds and exhibits lower particle emission during printing [89]. PLA prints have wider application than HIPS due to biocompatibility and susceptibility to biodegradation, which are important in biomedical application and tissue engineering [90]. Moreover, the price of 1 kg of PLA filament is comparable to that for HIPS. This is why PLA can be a good alternative to HIPS in rapid manufacturing of packaging prototypes using 3D printing technology [91].

### 3.3. Medical Application/Drug Delivery

Medical plastic has to be biocompatible, stable under different sterilization conditions, and robust to surface modification. While PLA fulfils all these requirements, PS is not applicable because of the cancerogenic properties of styrene and its very moderate biocompatibility [54]. However, there are several studies on improving the biocompatibility of PS, e.g., by nonequilibrium gaseous plasma treatment [92]. Both polymers can be sterilized by ethylene oxide, gamma radiation, and electron-beam radiation, however, due to the presence of a benzene ring in its structure, PS is more resistant to high radiation doses than PLA. PLA exhibits strong resistance to sterilization processes with use of an autoclave or dry heat [93]; standard PS is not autoclavable, but syndiotactic PS is excellent [94]. The main application of PS in the laboratory field is the production of different containers for a variety of liquids, cells, and bacteria, together with microspheres used as drug carriers and magnetic particles. The biocompatibility of PLA makes this material an excellent application as scaffolds for bone regeneration, implants, stents, along with bioresorbable surgical and orthopedic threads and dental implants. Owing to the good mechanical properties of PLA, it can be used in catheters, heart pumps, and epidural trays to replace PS [54]. PLA and PS are also used as a surface for adhesion and proliferation of fibroblast and osteoblast cell lines [95].

PLA is a promising bioplastic with mechanical properties comparable to those of PS. In addition to its established position as a material for biomedical applications, it can replace mass production plastics from petroleum. However, there are still challenges that need to be addressed, e.g., improvement of barrier properties, which play a very important role in maintaining food quality and safety [96]. Moreover, the cost of PLA manufacturing is still too high to compete with PS. That is why there is a need to find low-cost substrates and high-performance microorganisms to increase the efficiency of LA production and obtain low-cost, high-quality PLA. Another concern is the recycling of PLA. PLA can be easily degraded in the natural environment or in compost; however, the idea of introducing a large amount of waste for biodegradation is unreasonable and its transformation into chemical products more valuable than simply carbon dioxide and water should be considered. Currently, several attempts of PLA recycling have been made but an industrially feasible chemical recycling concept, in adherence to the fundamental principles of closed-loop recycling within a Circular Economy, has not yet been developed [97]. Other than PLA, products made from PS can be recycled, but the high cost of the recycling process and the segregation problem make the technology inefficient. Moreover, the production of biopolymers is considered more sustainable than petroleum-based materials due to the reduced net carbon footprint [98].

## 4. Polyethylene Terephthalate vs. Chitosan

The abbreviation “PET” is well-known to all consumers worldwide and stands for a petroleum-based synthetic polymer of terephthalic acid and ethylene glycol, i.e., polyethylene terephthalate (Figure 5). The history of this polymer production dates back to the 1940s when Whinfield and Dickson [99] patented terephthalic esters in the form of linear polymers. The second most important patent relating to the described plastic is the invention of Wyeth and Roseveare [100], i.e., a plastic soda bottle. In recent years, the global production of this plastic exceeded 30 million metric tons and a 4% market growth rate is expected in the coming years [101].

The technology of PET production has been developed over the years, and now the substrates for the synthesis of this polymer, i.e., terephthalic acid or dimethyl terephthalate and ethylene glycol, are obtained from fossil-based resources. Consequently, the first two compounds are produced from *p*-xylene and the diol is made through the oxidation of ethene. One of the major industrial-scale synthesis methods of polyalkylene terephthalates is the so-called melt polycondensation, including a two-step process in the presence of catalysts for (trans)esterification of substrates in an inert atmosphere at 460–500 K, and subsequently, polycondensation of the intermediates at reduced pressure and increased temperature (520–550 K) until obtaining enough viscous mixture resulting in the end product [102].

The technology affects the appearance and the final form of PET, wherein the amorphous state the polymer is transparent, while in the semicrystalline state PET appears opaque [103]. According to the ASTM (American Society for Testing and Materials) International Resin Identification Coding System the number 1 was assigned to PET, meaning low production costs together with high recycling possibilities [104,105]. PET materials are characterized by high strength, rigidity, and hardness. High thermal stability is observed, which is connected to the presence of para-substituted aromatic rings in the polymer structure. Such polyesters have a melting point above 520 K, and even a theoretical value of 565 K was calculated for PET if an accurate process of annealing would be used [103]. Moreover, both the T_g_ and density are strictly dependent on the final form of the polymer; hence, T_g_ for the amorphous state is 340 K and for the semicrystalline PET is about 350 K [104]. Amorphous PET has a density of 1.30–1.34 g/cm^3^ and semicrystalline PET is about 1.50 g/cm^3^. Crystalline PET is much denser than the amorphous form because polymer chains of the former are closely packed and parallel, while they are disordered in the latter [106]. Interestingly, exceeding the T_g_ point, thus the temperature above 340–350 K, may lead to the decreased resistance of PET to hydrolysis [107].

Polyethylene terephthalate has found its application in single-use food packaging, mainly for beverage bottles, but another major market is the textile industry and the production of clothes, shoes, and carpet fibers [101,108]. Features such as transparency, lightness, strength, and durability made PET bottles favored over those made of glass, and the results of the Coca-Cola Company life-cycle assessment study in 1969 revealed that bottles obtained from PET affected the environment less than their glass analogues [104]. The possibility of using PET in packaging material for food or pharmaceuticals results from low permeability for gases and solvents, together with low moisture absorption [109]. Chemical stability of polymers is also extremely important. PET’s stability is observed in weak acids and is also inert to several organic solvents from the alcohol, halogen, and ketone groups [104,105]. Strong acids, bases, hydrocarbons, and aromatic compounds are examples of chemicals that influence the stability of PET, and moreover, according to Lepoittevin and Roger [103], PET may be soluble in a mixture of phenol and trichloroethane, and in trifluoroacetic acid, *o*-chlorophenol, or hexafluoroisopropanol.

It is estimated that the degradation of plastic bottles lasts around 450 years, and 8% of solid waste weight is contributed to PET [110]. Owing to the fact that almost 60% of PET is discarded into landfills, solutions are being sought to reuse this plastic. Approaches such as bottle reuse systems, bans on plastic bags, and the introduction of taxes and deposit systems have been successfully applied in many geographical regions, e.g., Europe, America, and South Asia [111]. Currently PET waste is subjected to recycling processes using both mechanical and chemical methods. Reactive extrusion is often used because of its simplicity and the multiple extrusions carried out at 540–550 K lead to a decrease in molecular mass of the polymer [110]. Among chemical recycling methods, the following reactions are distinguished: hydrolysis at higher temperature and pressure, methanolysis, glycolysis, aminolysis, and ammonolysis, which lead to the depolymerization of PET. Products obtained from chemical depolymerization found their application in various industries, e.g., as cement replacement, corrosion inhibitors, paints, etc. [112]. Unfortunately, chemical recycling of polymers seems to be unprofitable because the polymerization of fossil-based substrates is much cheaper than reprocessing of polyesters [108].

PET is considered resistant to hydrolysis and enzymatic treatment, but the current prospects for its biodegradation are very promising. Numerous studies have shown the usefulness of bacteria of the genera *Ideonella*, *Bacillus,* and *Streptomyces*, along with the enzymes produced from the esterase and cutinase group, the so-called PETases [101,104,108]. The discovery of such microorganisms with their ability to produce PET hydrolyzing enzymes, and their possible application in polymer biodegradation may be an innovative approach in waste management in line with the circular economy system, which meets the goals of sustainable development [108].

Currently, new packaging materials, mainly those which are bio-based and biodegradable, are gaining interest among the consumers and food manufacturers. An interesting example of such a polymer is chitosan (Figure 6).

Chitosan is a biopolymer obtained after chitin deacetylation, the discovery of which is attributed to the French physiologist Charles Rouget, who in 1859 heated chitin (a glycan consisting of *N*-acetyl-D-glucosamine molecules) in an alkaline solution. From the chemical point of view, this polysaccharide is a linear polymer consisting of D-glucosamine units linked with β 1–4 glycosidic bonds. The process of removing acetyl groups is not often entirely performed; therefore there are several chitosan preparations available on the market with different degrees of deacetylation [113,114]. Interestingly, the industrial production of chitosan started in Japan in 1971 [115].

It is believed that chitin, along with cellulose, is one of the most common biopolymers found worldwide. The main sources of chitin and chitosan are crustaceans (shrimps and prawns, krills, or crabs), and insects, together with microorganisms, mainly fungi, algae, and some yeasts [113,116]. Shells and other inedible parts, i.e., crustacean waste, are a good source for these polysaccharides, especially since their contents in this arthropod taxon reach up to 20% dry weight and seafood consumption will grow increasingly [113,117]. In the last few years, several biotechnology companies, such as the Mycodev Group (Fredericton, NB, Canada), Chibio Biotech Co., Ltd. (Qingdao City, Shandong Province, China), and KitoZyme (Herstal, Belgium), have launched and have been producing the so-called “vegetal chitosan” or “mycochitosan”, a fully nonanimal source for this polymer obtained through the fermentation process with the use of filamentous fungi; hence, these preparations may be an alternative for people with shellfish allergies and for vegetarians [113].

The properties of chitosan and PET differ, especially in terms of strength and hardness. Therefore, these polymers cannot be replaced one-to-one, but other attributes of the former allow the use of plastics to be limited, as mentioned below. One of the main advantages of this polysaccharide is that it can be consumed as opposed to petroleum-based synthetic polymers. Chitosan has been considered as material for food packaging, but also as a dietary supplement, biofertilizer, and biopesticide. Moreover, hydrogels and wound-healing bandages based on chitosan are currently obtainable on the market [117]. This polymer could be used for bodyweight reduction, but there are some concerns that it may interact with fat-soluble vitamins. The lowest observed adverse effect level (LOAEL) for chitosan is relatively high and it accounts for 450 mg/kg for men and 6000 mg/kg for women [117]. In the case of oral median lethal dose (LD_50_) in mice, it was revealed that the value was higher than that of sucrose and it exceeded 16 g/day/kg body mass [114].

Furthermore, its potential widespread application comes from both biological and physical properties, such as antimicrobial activity, biodegradability, and film-forming ability. Chitosan is not soluble in water and organic solvents but dissolves in dilute acetic or hydrochloric acids. Chitosan-based edible coatings and films are able to extend the shelf life of food products, prolong their quality, improve nutritional and antioxidant properties, and prevent the growth of food-spoilage microorganisms. In order to improve some features of such films, various additives are added, such as plasticizers in the form of polyols (e.g., glycerol) and emulsifiers. Chitosan can also be blended with other biopolymers, such as alginate, pectin, starch, or caseinate, or enriched with different valuable compounds (e.g., polyphenol extracts) to enhance its applicability [118].

Among various polymers, chitosan is distinguished by its antimicrobial activity. Interestingly, two different mechanisms may be responsible for this activity, but in each case the chemical structure is the clue. The first mechanism is associated with chitosan binding to DNA molecules causing bacterial cell death. In the second one, chitosan and more specifically the amino groups in the chain of this polysaccharide in the acidic medium generate a cationic charge that binds to negatively charged bacterial cell walls and membranes, resulting in permeability disturbances and cell leakage. In addition, by lowering the pH value, the improvement in the antibacterial activity is observed due to the increase in the protonation of the amino groups [119].

Chitosan shows remarkable potential for biomedical applications. Apart from already-mentioned bioactive dressings for healing wounds and burns, this polysaccharide can be used in preparing drug delivery systems of different forms (tablets, gels, granules, films, or microcapsules) or can serve as an artificial kidney membrane impermeable for serum proteins, and potential material for contact lenses, specifically due to its good tolerance by living tissues [116]. Over the past few decades, scientists were also interested in the use of chitosan in tissue engineering and regenerative medicine. Both PET and expanded polytetrafluoroethylene (ePTFE) are usually standard materials for prosthetic vascular grafts, but the need for small-diameter (<4 mm) application becomes problematic with the use of these two synthetic polymers [120]. Chupa et al. [121] indicated the meaningful potential of chitosan and its complexes with glycosaminoglycans or dextran sulfate in the preparation of newly bioactive materials that exhibited activity both in vitro and in vivo and modulated the activity of smooth muscle and vascular endothelial cells.

Pure chitosan films are not thermoplastic and thus cannot be softened by heating, thus their extrusion, molding, or heat-sealing is limited [122]. Some properties of chitosan-based films, e.g., water vapor and oxygen permeabilities, are thus important parameters influencing the applicability of the coating or film for food packaging (since the moisture and oxygen levels can lead to lowering the quality of food), can change depending on the addition of some materials incorporated into chitosan films, as well as the characteristics of chitosan, i.e., the degree of deacetylation and the methodology of coatings and films manufacturing. Wang et al. [118] reported that the use of organic compounds such as polyphenols in chitosan films led to a decrease in water vapor permeability due to the interactions between compounds and limiting interactions with water molecules. Moreover, the use of nanoparticles led to similar observations because good dispersion in the film and filling the spaces resulted in hindered migration of water. Furthermore, it is believed that the tightly packed structure of chitosan with plenty of hydrogen bonds may have limited oxygen permeability, and the incorporation of chitosan-based films with graphene oxide nanosheets or silver nanoparticles caused a significant reduction in oxygen permeability [119].

In the near future, chitosan-based packaging may find its niche in producing active and intelligent packaging. The first type, in addition to the features typically assigned to packaging, is characterized by carrying additional properties maintaining the quality and increasing the safety of the products, which in the case of chitosan relates to its antimicrobial and antioxidant activities [119]. The term “intelligent” in the case of the latter refers to the possibility of using packaging for real-time food quality monitoring. This approach arouses the interest of scientists, thereupon extensive research is undertaken, but also industry attention is attracted. Recently colorimetric and pH-indicating films are under special examination, where both their manufacturing and specific properties on the selected food products are assessed. Methylene blue is an example of a dye applied in chitosan-based films, whereby its color depends on oxygen concentration in the atmosphere surrounding the product and decreasing oxygen content may indicate microbiological contamination [118]. As pH-indicating compounds, both synthetic and natural substances are used in chitosan films. The studies on the use of alizarin, along with anthocyanins from purple potato or grapes were conducted and confirmed the applicability of these compounds to monitor changes in food, where pH alternations may suggest that the product is stale or contaminated [118,119].

Concluding, it cannot be clearly stated which polymer is better, PET or chitosan. Similarly it cannot be confirmed with certainty that the latter, i.e., a polymer of natural origin, will replace PET because they differ significantly in their properties. PET is primarily a packaging material used in the beverage industry with high hardness and rigidity. Inversely, chitosan is used to create coatings and films, and the possibility of producing edible packaging is an additional advantage. Finally, the prospect of producing active and intelligent packaging based on chitosan, which will be also edible, biodegradable, and compostable, points in favor of its use. Therefore, the production of chitosan-based packaging will be a limitation in the use of PET, replacing it in selected applications, rather than a complete replacement.

## 5. Polyvinyl Chloride vs. Pullulan

Polyvinyl chloride—PVC—is a long-chain thermoplastic polymer produced by a free-radical polymerization of vinyl chloride monomer. Industrial synthesis of PVC is dated to the early 1930s, and recent estimates account production volume as third among plastics after polypropylene and polyethylene, and up to 25% of total plastic production [21]. The basic raw materials for the PVC synthesis come from crude oil and sodium chloride and hence only 43% of this polymer mass is petroleum-based [123]. The chemical formula of PVC is (C_2_H_3_Cl)_n_, and the chemical structure is a long chain with the repeating unit of vinyl chloride, as shown in Figure 7.

As a thermoplastic material with excellent chemical and mechanical properties, PVC has widespread uses. These properties together with low production cost affect the economic significance of PVC worldwide. The most important properties are polymer durability, high chemical resistance, resistance to water and weather conditions, and adhesiveness [124,125]. Moreover, because of its high polarity, it has the ability to accommodate a wide range of additives such as stabilizers, plasticizers, lubricants, or pigments [126]. The addition of different chemicals to PVC resins modifies its properties and may change the possible way of use. The unplasticized PVC is hard and rigid and can be used in plumbing, construction, fencing, or drainage systems, whereas plasticizers incorporated into the polymer make it softer and more flexible, thus useful in electrical cable insulation, medical devices, or inflatable toys [127]. These hard and soft variants of PVC compounds differ considerably in terms of the T_g_ and flexibility at specific temperatures [128]. Additionally, high transparency of this polymer is useful in film production or light-transmitting panels. It is a lightweight material, with high strength-to-weight ratio, which imparts no taint or taste. Notwithstanding, almost 70% of PVC compounds is used in the construction industry [129], next in medicine (flexible blood containers or inhalation masks) [130,131], and the packaging industry (wrap films with good oxygen barrier properties) [132].

Despite the outstanding position of PVC mainly among medical polymers, it is considered to be harmful to both human and environment because of various chemicals and dangerous degradation products released during its life cycle [123]. Its complex composition together with low thermal stability makes this polymer difficult to recycle, but there are available techniques for the management of PVC waste, both mechanical [133] and feedstock [134].

Pullulan is an extra-cellular, unbranched, water soluble, neutral, nonionic, nontoxic, nonmutagenic, noncarcinogenic exopolisaccharide. This biopolymer is obtained from fermentation medium of the fungus-like yeast *Aureobasidium pullulans,* generally referred to as “black yeast”. The final yield of pullulan is highly affected by media composition and culture conditions [135,136]. The large scale production started in 1976 by Hayashibara Co., Ltd. (Okayama, Japan), which is still the leading commercial producer worldwide together with Shandong Jinmei Biotechnology Co., Ltd. (Zhucheng, China). Pullulan is generally marketed as white or off-white dry powder or capsules [137]. The chemical formula of pullulan has been suggested to be (C_6_H_10_O_5_)_n_, and its chemical structure consists of repeating units of maltotriose linked with each other by the α 1–6 glycosidic bonds [138,139,140]. Pullulan structure is often seen as an intermediate between amylose and dextran structures. The chemical structure is shown in Figure 8.

The peculiar structure of pullulan (mainly the α 1–6 linkages) affects strongly its high solubility in both cold and hot water, and its lack of ability to form gels. It is responsible for the high structural flexibility, but is also reflected in the lack of crystalline regions within the polymer, which has a completely amorphous organization [141]. Pullulan compounds have a high heat resistance and are biodegradable in biologically active environments, therefore it can be utilized in many different ways. It has a significant mechanical strength, adhesiveness, thick film, and fiber formability, stability of aqueous solution over a broad range of pH, low viscosity, and good oxygen- and moisture-barrier properties [136,142]; it also inhibits fungal growth in food [143]. Pullulan can be formed into compression moldings that can resemble PS and PVC in transparency, gloss, hardness, and strength, but which are far more elastic [135]. Additionally, the capacity to form thin layers, nanoparticles, flexible coatings, and standalone films means it can successfully replace other synthetic polymers derived from petroleum, such as polyvinyl alcohol [138].

This polysaccharide is also colorless, tasteless, odorless, and, what is more important, edible, however, not attacked by the digestive enzymes in human gut [136,142]. It has a “generally recognized as safe” status in the United States [144] and in the European Union is a food additive of microbial origin E1204 [145].

As a nonpolluting “plastic”, the biodegradable and biocompatible biopolymer pullulan could be used in different sectors, especially pharmaceutical (hard and soft capsules, drug delivery systems, anticancer nanoparticles) [146,147], biomedical (wound healing) [137], environmental, food, and cosmetics (body and skin application) [148]. However, despite all the pullulan advantages, its high cost is the limiting factor for wide scale applications.

The most emerging area of interest in pullulan application is still the food packaging sector. Because of the nonbiodegradable character of synthetic polymers and their great impact on the environment, research remain focused on developing ecofriendly and biodegradable food packaging systems obtained from natural sources. Such applications are edible coatings and active films, which are known to protect food, extend the shelf life of the product, and improve its quality. A thin layer of pullulan is formed directly on the surface of a product and can be safely eaten with a protected food. Pullulan was used as an edible coating on, e.g., highbush blueberries [149], Fuji apples [150], and white asparagus [151]. Moreover, pullulan is also an excellent medium for different compounds, which play an important role in prolonging shelf life: chitosan [152], thymol [153], pectin [154], or propolis [142].

## 6. Summary

The most important properties of the polymers discussed earlier in this paper have been collected and are presented below (Table 1).

The total replacement of crude oil-based polymers with bio-based polymers is not yet possible, mainly because of functional properties of synthetic polymers, namely durability, mechanical, and water vapor barrier. Therefore, to partially eliminate synthetic polymers from packaging systems and to reduce environmental impact, biopolymers combined with crude oil-based polymers are of interest. Chemical modification helps in combining specific characteristics of different polymers, resulting in a unique and suitable material with interesting properties [161].

## 7. Conclusions

Bio-based polymers can replace crude oil-based polymers in a variety of applications from everyday products to medical materials or advanced technology. Yet a noticeable difference in the physicochemical parameters between both groups of polymers prevents the petroleum-based synthetic polymers from being substituted directly by their bio-based counterparts. Nevertheless, the replacement of crude oil-based polymers by bio-based polymers in only selected applications would considerably diminish the net carbon footprint and create sustainable solutions in polymeric materials management. Many bio-based polymers such as PHB and PLA, which are commonly adapted in the production of biocompatible devices extensively used in surgery, drug delivery, and cardiovascular systems, or dentistry, can successfully compete with their hydrocarbon polymeric analogues, namely PE and PS. PLA meets all the requirements to be a perfect biomaterial, whereas PS, which may be contaminated with residual styrene, might be considered cancerogenic. Similarly, PVC, used primarily in the construction, medical, and packaging industry, is regarded as toxic to humans and the environment due to harmful compounds released during its life cycle.

The food packaging sector, which is dominated by PET, a polymer resistant to biodegradation, is currently searching for a bio-based and biodegradable replacement. Chitosan, one of the most widespread biopolymers found in nature, can at least in part rival PET and contribute to a considerable reduction in plastics. This polysaccharide, which has a film-forming ability, is edible and can be consumed together with the food product; additionally, its antimicrobial and antioxidant properties can increase the safety and stability of the protected foodstuff. Edible coatings formed by pullulan are reported to prolong the shelf-life of numerous fruit and vegetables; these properties can be enhanced by the addition of other bioactive molecules. All of these ecofriendly and innovative advantages may constitute a springboard for creating active and intelligent packaging in the future. However, PET is still the best option for the beverage industry, which needs hard and rigid polymeric materials for packaging.

The implementation of bio-based polymers in many industrial sectors is currently limited mainly due to the high costs of their production. The manufacturing of crude oil-based polymers remains still more cost-effective, although such polymers are resistant to biodegradation, and their recycling is as yet uneconomical.

A significant challenge for the future is to synthesize novel, sustainable bio-based polymers with such functionalities that could enable the substitution of their conventional analogues. Biotechnology is a strategic factor, which can notably contribute to the transition from fossil-derived plastics to bioplastics acquired from renewable resources. Technological advancements and new biotechnological findings may allow for better development of bio-based materials and reduce production expenditures.

## Figures and Tables

**Figure 1 polymers-14-05551-f001:**
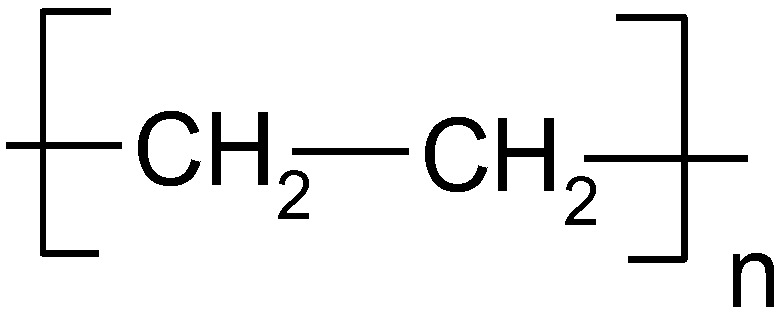
The fragment of polyethylene structure.

**Figure 2 polymers-14-05551-f002:**
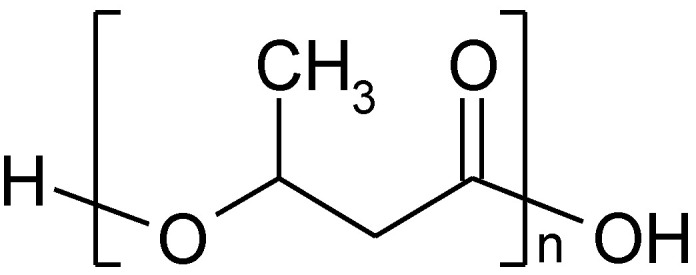
The fragment of polyhydroxybutyrate structure.

**Figure 3 polymers-14-05551-f003:**
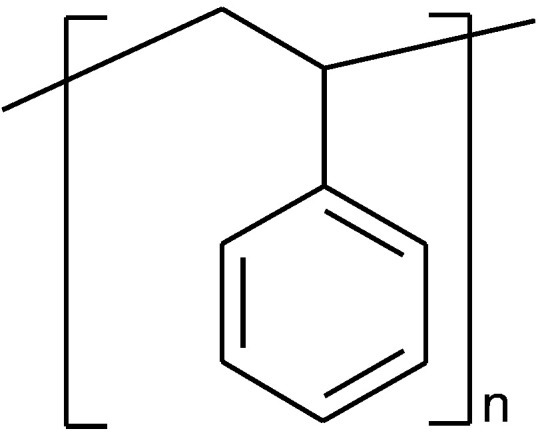
The fragment of polystyrene structure.

**Figure 4 polymers-14-05551-f004:**
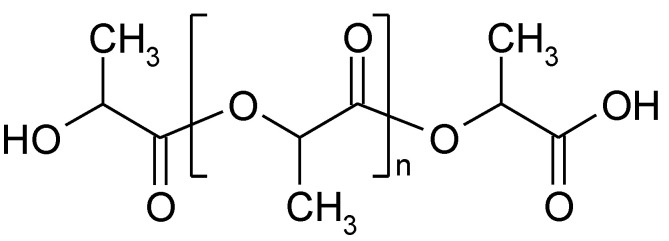
The fragment of polylactide structure.

**Figure 5 polymers-14-05551-f005:**
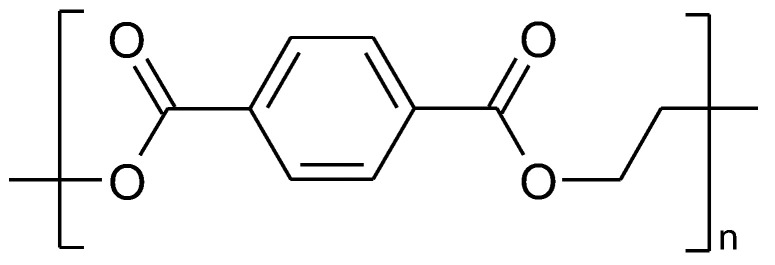
The fragment of polyethylene terephthalate structure.

**Figure 6 polymers-14-05551-f006:**
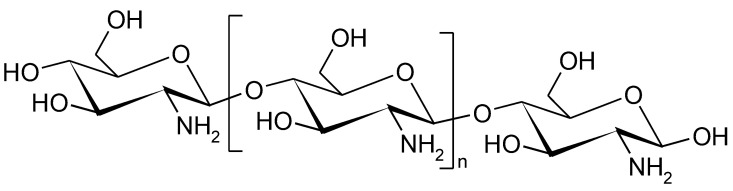
The fragment of chitosan structure.

**Figure 7 polymers-14-05551-f007:**
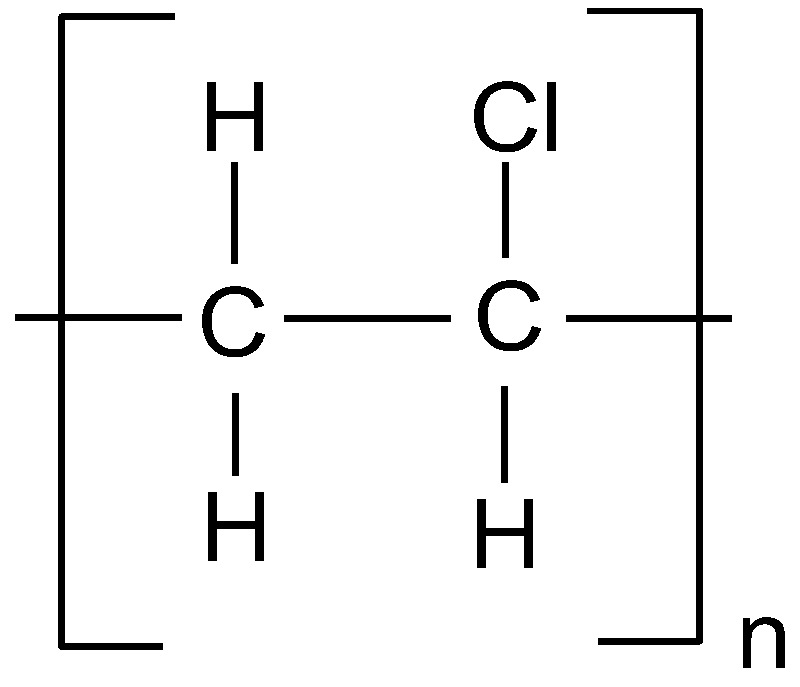
The fragment of polyvinyl chloride structure.

**Figure 8 polymers-14-05551-f008:**
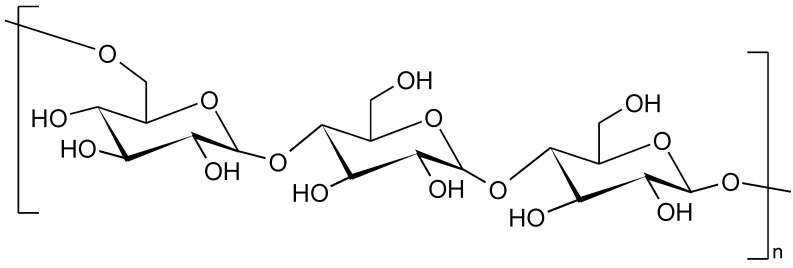
The fragment of pullulan structure.

**Table 1 polymers-14-05551-t001:** Selected properties of all the polymers discussed in this paper.

Property	Pairs of Polymers (Fossil–Bio)							
	PE–PHB		PS–PLA		PET–Chitosan		PVC–Pullulan	
Density (kg/m^3^)	950 ^c^	1262 ^d^	1111–1127 ^d^	1248–1290 ^d^	1300–1500 ^c^	1000 ^e^	1330–1380 ^c^	1500 ± 100 ^e^
Young’s Modulus (GPa)	0.7 ^c^	3–3.5 ^a^	3.4 ^c^	3.0 ^f^	1.7 ^d^	0.63 ^g^	3.4 ^c^	N/A *
Tensile Strength (MPa)	30–40 ^a^	20–40 ^a^	34.5–68.9 ^b^	50–70 ^f^	62 ^a^	9.7 ^g^	48 ^c^	40.2 ^e^
Elongation at Break (%)	200–700 ^a^	5–10 ^a^	1–2.3 ^b^	4.0 ^f^	30–80 ^b^	2.6 ^g^	85–104 ^d^	N/A *
Flexural Strength (MPa)	40 ^d^	33–40 ^d^	68.9–103 ^b^	100 ^f^	96.5–124.1 ^b^	N/A *	72 ^d^	N/A *
Izod Notched Impact Strength (J/m)	60–80 ^d^	35–50 ^d^	21 ^b^	26 ^b^	59 ^b^	N/A *	21–53 ^d^	N/A *
Degree of Crystallinity (%)	25–80 ^a^	50–60 ^a^	N/A *	3.5–14 ^a^	7.97 ^a^	57 ^h^	11–15 ^d^	N/A *
Melting Temp. (K)	390–410 ^a^	440–450 ^a^	513 ^d^	440–470 ^a^	530 ^a^	360 ^e^	485–583 ^d^	193 ^e^
Glass Transition Temp. (K)	140–370 ^a^	278–282 ^a^	373 ^d^	320–330 ^a^	340–354 ^a^	N/A *	353–370 ^d^	N/A *

^a^ [29], ^b^ [53], ^c^ [155], ^d^ [156], ^e^ [157], ^f^ [158], ^g^ [159], ^h^ [160], * N/A—not available.

## Data Availability

Not applicable.

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
