# Peer review of "Chemical Structures, Properties, and Applications of Selected Crude Oil-Based and Bio-Based Polymers"

_polymers, 2022, doi:10.3390/polym14245551_

Round 1
Reviewer 1 Report
Review does not have sufficient information for the possible publication.
Very few properties and monomers are discussed.
Reviewer 2 Report
The study of issues related to the production and use of bio-polymers as a replacement for traditional polymers obtained from oil is an important and actual task. But, as a rule, a more realistic scenario for such replacement is the search for renewable sources of raw materials to isolate monomers from them that have a similar structure to traditionally used monomers and obtain materials similar in properties to traditional polymers. The issue of biodegradability of such polymers is secondary, since the way of recycling looks more promising, as the Authors also mentioned in their manuscript.
In this regard, the comparison of a number of oil-based and bio-based polymers carried out in this manuscript looks somewhat meaningless: the main areas of use and the corresponding advantages for PE, PS, PET, PVC on the one hand, and PHB, PLA, chitosan, pullulan on the other hand, are practically do not match. Therefore, they cannot be adequately substituted for each other. And, if a direct comparison of PE-PHB and PS-PLA is still justified to some extent in terms of similar performance characteristics, then PET-chitosan and PVC-pullulan pairs were chosen for comparison very unsuccessfully, in my opinion, because these polymer pairs have very different characteristics and corresponding applications.
Instead of comparing and listing the characteristics of well-known and repeatedly described polymers (PE, PS, PET, PVC), it might be more useful to take a more extensive and in-depth review of only polymers derived from renewable raw materials and / or biodegradable.
Nevertheless, the text of the manuscript itself is well-written and contains a number of useful literature data on the properties and applications of polymers. In general, despite the low scientific importance, this manuscript may be published at the decision of the Editor.
I would also like to point out a few errors in the text of the manuscript:
1) on line 99 there is an error in the writing of the name (correct - Carothers);
2) on lines 138-139 it is not entirely clear what the authors mean by the phrase “each having a different density range - and; ".
Reviewer 3 Report
In the current review data on selected popular crude oil-based and bio-based polymers has been collected in order to compare their practical applications resulting from their composition, chemical structure, and related physical and chemical properties.
The review is well organized and interesting. Some minor changes are required before publication
1. Abstract part should be improved
2. The strength of the current review should be highlighted in the introduction part.
3. Figures given herein are not so significant. Authors are invited to include more interesting mechanisms to support such data.
4. Table 1 needs improvement. It should be better organized
5. Conclusion is too short
6. Some references should be updated
Reviewer 4 Report
This paper collects some data on oil-based and bio-based polymers for comparison and points out possible future prospects for bio-based polymers. The hierarchical expression of this paper is clear and has a certain logical coherence. However, in my opinion, the manuscript required minor revision before considering for publication:
The conclusion should be based on the discussion of the content of the article.
Furthermore, there are some minor details in this manuscript:
In the part of 2. Polyethylene vs. Polyhydroxybutyrate, please increase the research on the degradation of PHB. Please supplement the relevant references for the findings of PE.
In line 62, suddenly adds a third group without mentioning what the first and second are. Please explain the first two group.
Line 624 should be written according to the format of the text. Give proper explanation to Table 1.
There are some grammatical errors in the writing, please correct them.
Round 2
Reviewer 1 Report
The authors replies are satisfactory. The manuscript can be accepted for possible publication.